# Automatic Detection and Segmentation for Group-Housed Pigs Based on PigMS R-CNN [note 1]

**DOI:** 10.3390/s21093251

**Published:** 2021-05-07

**Authors:** Shuqin Tu, Weijun Yuan, Yun Liang, Fan Wang, Hua Wan

**Affiliations:** College of Mathematics and Informatics, South China Agricultural University, Guangzhou 510642, China; Tsq5_6@scau.edu.cn (S.T.); ywj@stu.scau.edu.cn (W.Y.); 201827010518@stu.scau.edu.cn (F.W.); wanhua@scau.edu.cn (H.W.)

**Keywords:** pig identification, mask scoring R-CNN, soft-NMS, group-housed pigs

## Abstract

Instance segmentation is an accurate and reliable method to segment adhesive pigs’ images, and is critical for providing health and welfare information on individual pigs, such as body condition score, live weight, and activity behaviors in group-housed pig environments. In this paper, a PigMS R-CNN framework based on mask scoring R-CNN (MS R-CNN) is explored to segment adhesive pig areas in group-pig images, to separate the identification and location of group-housed pigs. The PigMS R-CNN consists of three processes. First, a residual network of 101-layers, combined with the feature pyramid network (FPN), is used as a feature extraction network to obtain feature maps for input images. Then, according to these feature maps, the region candidate network generates the regions of interest (RoIs). Finally, for each RoI, we can obtain the location, classification, and segmentation results of detected pigs through the regression and category, and mask three branches from the PigMS R-CNN head network. To avoid target pigs being missed and error detections in overlapping or stuck areas of group-housed pigs, the PigMS R-CNN framework uses soft non-maximum suppression (soft-NMS) by replacing the traditional NMS to conduct post-processing selected operation of pigs. The MS R-CNN framework with traditional NMS obtains results with an F1 of 0.9228. By setting the soft-NMS threshold to 0.7 on PigMS R-CNN, detection of the target pigs achieves an F1 of 0.9374. The work explores a new instance segmentation method for adhesive group-housed pig images, which provides valuable exploration for vision-based, real-time automatic pig monitoring and welfare evaluation.

## 1. Introduction

With the development of artificial intelligence and automation technology, utilizing video cameras to monitor the health and welfare of pigs has become more important in the modern pig industry. In group-housed environments, instance segmentation of pigs includes detection, which automatically obtains the positions of all pigs, and segmentation, which distinguishes each pig in the images [1]. Many high-level and intelligent pig farming applications, such as pig weight estimation [2], pig tracking [3], and behavior recognition [4,5,6,7], require accurate detection and segmentation of pig objects in complex backgrounds. The premise and foundation of pig behavior analysis involves the accurate detection and segmentation of group pig images [8]. Therefore, detecting and segmenting group-housed pigs can help improve the efficiency of instance segmentation, complete high-level applications, and improve the welfare of pigs in pig farms. 

Digital image processing combined with pattern recognition studies use automatic detection and segmentation techniques to group-housed pigs [9]. Pig detection and segmentation methods include two categories based on non-deep learning and deep learning algorithms. These non-deep learning approaches have more mature technologies and have been widely applied in video surveillance of pigs [1,10,11,12]. Guo et al. [10] proposed a pig’s foreground detection method based on the combination of a mixture of Gaussians and threshold segmentation. This approach achieved an average pig object detection rate with approximately 92% in complex scenes. Guo et al. [11] also proposed an effective method for identifying individual group-housed pigs from a feeder and a drinker using a multilevel threshold segmentation. The method achieved a 92.5% average detection rate on test video data. An approach [5] was proposed to detect mounting events amongst pigs for pig video files. The method used the Euclidean distances of the different parts of pigs to automatically recognize a mounting event. This approach can obtain the results of sensitivity, specificity, and accuracy, with 94.5%, 88.6%, and 92.7%, respectively, for identifying mounting events. The algorithm [12] for the group-housed pig detection was developed, and it used Gabor and LBP features for feature extraction and classified each pig by SVM. A recognition rate with 91.86% was obtained by this algorithm. Li et al. [1] combined appearance features and the template matching framework to detect each pig under a group-housed environment. From these related studies, we found that these non-deep learning methods have some disadvantages, and that feature extraction and recognition of the pigs are separated. Meanwhile, feature extractions of these methods obtain features with handed-design features and cannot automatically learn features from the amount of data, which causes these methods to lack robustness.

The studies on pig detection and segmentation methods, based on deep learning models, have increased in recent years, since Girshick et al. [13] proposed the R-CNN method. These algorithms can automatically extract the pig’s target features in an image and avoid the process of extracting the target features by artificial observation, so the obtained model has strong universality. Faster R-CNN [14] was used to detect pig targets and classify lactating sow postures, including standing, sitting, sternal recumbency, ventral recumbency, and lateral recumbency, using depth images [15]. Yang et al. [16] first used faster R-CNN to detect individual pigs and their heads. Then, a behavior identification algorithm was implemented for feeding behavior recognition from a group-housed pen. Finally, the results of a precision rate with 0.99 and recall rate with 0.8693 can be obtained for feeding behavior recognition of pigs. Xue et al. [17] proposed an approach based on a fully convolutional network (FCN) [18] and Otsu’s thresholding to segment the lactating sow images. The approach achieved a 96.6% mean accuracy rate. He et al. [19] proposed an end-to-end framework named mask R-CNN for object detection and segmentation. The mask R-CNN achieved good results for the challenging instance segmentation dataset COCO [13], and was used in cattle segmentation [20]. These recent research results show that pig detection and segmentation approaches, based on deep learning models, were effective and widely used under complex scene environments. However, when pigs are under heavy overlap and adhesion, pig instance segmentation methods, based on non-deep learning and deep learning algorithms, have some difficulty accurately detecting individual pigs with a low miss rate. 

Segmenting the adhesive pig images is important to the next extraction of pig herd behavioral characteristic parameters. The adhesion segmentation methods mainly include ellipse fitting, watershed transformation, and concave point analysis. In [4,9], the least square method was used to conduct ellipse fitting for pigs, and then it separated the adhered pig bodies according to the length of the major and minor axis, and the position of the center point, but this method did not evaluate the performance of the ellipse fitting method when several pigs adhered. Xiong et al. [21] divided piglet adhesion into four cases, as follows: no adhesion, slight, mild, and severe adhesion. Firstly, the contour line of the adhered part was extracted. Then, the contour line, according to the concave point, was segmented. The ellipse fitting for the contour lines was carried out after segmentation. Finally, five ellipse-screening rules were developed. The ellipses, which did not comply with the rules, were integrated. The recognition accuracy of this method for pigs was over 86%. A Kinect-based segmentation of touching-pigs was used for segmentation of touching-pigs, by applying YOLO and CPs [8], and this method was effective at separating touching-pigs with an accuracy of 91.96%. Mask scoring R-CNN (MS R-CNN) [22] was explored for instance segmentation of standing posture images of group-housed pigs, from the top view and front view pig video sequences, which achieved a best F1 score with 0.9405 on pig test datasets in our previous work [23]. However, severe adhesion was not analyzed, but is further researched in these studies.

Although these works have been performed effectively on pig detection and segmentation, most approaches are designed under fewer pigs/one pig environments or top view images; it is difficult for these algorithms to detect each pig in a group-housed condition when the pigs are closely grouped. Based on the previous work of our team [23], we propose a PigMS R-CNN model, which can efficiently obtain the instance segmentation mask for each pig, while simultaneously decrease the faulty detection results for group-housed pigs. To reduce the missed rate of pigs in group-housed environments, soft-NMS [24] was used in the PigMS R-CNN model. The algorithm gains improvements in precision measured over multiple overlap thresholds, which are especially suitable for group-housed pig detection. The PigMS R-CNN model, combining the MS R-CNN and soft-NMS network, is simple to implement and does not need extra resources. We can easily use the improved approach for a pig instance segmentation application. 

## 2. Materials and Methods

### 2.1. Data Acquisition

The experiment’s dataset was collected in “Lejiazhuang Pig Farm” of live pigs in Foshan city, Guangdong Province, China. The settings of the cameras of the experimental pigsty are shown in Figure 1. In Figure 1a, the pigsty was 3 m high, 7 m long, and 5 m wide. Figure 1b,c present the images, in video surveillance, of top view and front view. A camera was placed in the middle of the pigsty, 2 m above the ground. Another camera was placed in front of the pigsty, 1.35 m above the ground. We used the FL3-U3-88S2C-C camera, which obtained images at 1920 × 1080 pixels.

To generate sufficient images in the experiments, we focused on 5 pigsties, where the number of pigs included is 3–20; video data were randomly obtained during a course of 10 days, each day containing over 7 h of video, from 9 a.m. to 4 p.m. The video dataset was saved in AVI format and the frame frequency of the video was 25 fps. We chose 420 images as the dataset. Among 420 images, 290 images that included 2838 pigs were used for the training set, and 130 images that included 1147 pigs were used for the test set. We marked pigs manually with VIA software for these images. The typical augmentation methods, including left-right flipping, rotation, and re-scaling were automatically used to enlarge the training dataset. With no external light used, all images showed uneven illumination. Therefore, the dataset reflected the common characteristics of pig monitoring, and it can objectively estimate a pig’s instance segmentation performance in a group-housed condition. 

### 2.2. Data Labeling

To evaluate the proposed algorithm, the frames with the standing status of pigs in videos were labeled to illustrate the detection and segmentation results of the algorithm. VIA (http://www.robots.ox.ac.uk/~vgg/software/via/ accessed on 7 May 2021) is an open-source image annotation tool that can be used online or offline. VIA software can label rectangles, circles, ellipses, polygons, points, and lines, set area properties, and save the annotation information as CSV or JSON file formats. 

In this study, we used it to label the contour of the group-house pigs with the shape of the polygon area, extracted the annotation information of the pig’s contour, and saved as a JSON file format. These JSON files included image name, image size, label name, anchor coordinate information of each pig object in each image, and area attribute name. Labeling the 420 images with each, including the 3–20 pig objects, cost about 180 person-hours. In the 420 images, the total number of pigs was 3985, including some incomplete pigs (with over half of their bodies visible) in the image borders.

## 3. The Proposed Approach

### 3.1. The PigMS R-CNN Model

The PigMS R-CNN model (as shown in Figure 2) based on MS R-CNN [22] included three stages. In the first stage, a residual network of 101 (or 50) layers, combining the feature pyramid network (FPN), was used as a feature extraction network to obtain feature maps for input images. FPN can obtain different levels of the feature maps according to three scales. In the second stage, the region proposal network (RPN) extracted the regions of interest (RoIs), according to the feature maps. In the third stage, for each RoI, we can get the location, classification, and segmentation results of detected pig targets in the group-housed scenes through the category, and regression, and mask three branches in the PigMS R-CNN head network. In the segmentation branch, the Maskiou head in FCN was used to regress between the predicted mask and the true ground mask to improve the segmentation accuracy.

In addition, the third stage extracted features using RoIAlign from each candidate RoI, and performed BB regression, classification with softmax, and a binary mask prediction for each potentially detected pig by FCN. During the process of BB regression, NMS was applied to these potentially detected pigs to remove highly overlapping BB and obtain the ultimate locations of the pigs. The steps are detailed in the following sections. 

### 3.2. The Feature Extraction 

The feature extraction uses backbone architecture to extract different levels of features over an entire image. The backbone architecture includes two parts, named ResNet [25] and FPN [26]. The deep ResNets are easy to optimize for training and obtain high accuracy from greatly increased depth compared with other networks. ResNets with 50-layer, 101-layer, and 152-layer depths are the most commonly used residual structures on detection and segmentation. In this paper, with comprehensive consideration of accuracy and operation time, the ResNet-101 network of a depth of 101 layers was used in the implementation of the mask R-CNN, and primarily extracted features from the three convolutional layers of the third, fourth, and fifth stages, as shown in Figure 3, left, which fed to the next multiscale backbone architecture.

Another part of effective backbone architecture was FPN, proposed by Tsung Yi Lin [26]. FPN can construct a multi-output feature pyramid from a single-scale input by using top-down architecture and lateral connections, as shown in Figure 3, middle and right. The detailed process of Figure 3 is described in the following.

The conv1, conv2, conv3, conv4, and conv5 outputs are obtained from the last residual blocks in the ResNet101 network. First, the top-layer feature map (P5) can be achieved by performing a 1 × 1 convolutional layer on conv5. Then, the upsampled feature map was generated by a factor of 2 on P_5_, and the feature map P_4_ (Figure 3, right lateral connections) can be obtained by element-wise addition operation for the upsampled feature map, and the result, which is obtained by performing a 1 × 1 convolutional operation on $C_{4}$. Finally, according to $\left\{C_{3},C_{4},C_{5}\right\}$, this final set of feature maps is $\left\{P_{3},P_{4},P_{5}\right\}$.

The backbone architecture of PigMS R-CNN in this study uses ResNe-101+FPN. The feature maps of backbone architecture for a housed-pig image are shown in Figure 4. Figure 4a shows source image, Figure 4b–d extract low-level features, including texture and edge contour information. Figure 4e represents high-level abstract features.

### 3.3. RoIs Generation Based on RPN

The candidate RoIs will be produced by RPN using the feature maps from the first stage. The convolution layers of a pre-trained network are followed by a 3 × 3 convolutional layer in the RPN. The function of this operation is to map a large spatial window or receptive field in the input image to a low-dimensional feature vector at a window location. Then two 1 × 1 convolutional layers are used for classification and regression operations of all spatial windows [14].

In the RPN, the algorithm introduces anchors to manage different scales and aspect ratios of objects. An anchor is located at each sliding location of the convolutional maps, and lies at the center of each spatial window, which is associated with a scale and an aspect ratio. Following the default setting of [26], five scales (32^2^, 64^2^, 128^2^, 256^2^, and 512^2^ pixels) and three aspect ratios (1:1, 1:2, and 2:1) are used, and k = 15 anchors at each location are created and used for each sliding window. These anchors go through a classification layer (cls) and a regression layer (reg). The RPN then completes the following two tasks: (1) determining whether the anchors are targets or non-targets (2k); (2) performing coordinate correction on the target anchors (4k). In the classification layer branch, two scores (target and non-target) are generated for each anchor; in the regression layer branch, the parameterizations of the four coordinates are corrected, and shown as Equation (1). For each target anchor:(1)$$\begin{array}{l} m_{x}=(x-x_{a})/w_{a},m_{y}=(y-y_{a})/h_{a},m_{w}=\log(w/w_{a}),m_{h}=\log(h/h_{a}),\\ m_{{}_{x}}^{*}=(x^{*}-x_{a})/w_{a},m_{{}_{y}}^{*}=(y^{*}-y_{a})/h_{a},m_{{}_{w}}^{*}=\log(w^{*}/w_{a}),m_{{}_{h}}^{*}=\log(h^{*}/h_{a}), \end{array}$$
where x,y denote the two coordinates of the box center, w,h is the width and height of the box. The $x,x_{a},$ and $x^{*}$ variables denote the predicted box, anchor box, and ground-truth box respectively (likewise for y,w,h). Finally, after sorting scores of the target anchors in descending order, the first n anchors are selected for the next stage of detection.

The loss function of training RPN is as follows:(2)$$L\left(\left\{p_{i}\right\},\left\{t_{i}\right\}\right)=\frac{1}{Ncls}\overset{}{\underset{i}{\Sigma }}L_{cls}\left(p_{i},p_{i}^{*}\right)+\lambda \frac{1}{Nreg}\overset{}{\underset{i}{\Sigma }}p_{i}^{*}L_{reg}\left(t_{i},t_{i}^{*}\right)$$
where i is the index of an anchor and $p_{i}$ is the predicted probability of anchor i, which is taken as a positive object. If the anchor is positive sample, the ground-truth label $p_{i}^{*}$ is 1, otherwise it is 0. The $m_{i}$ vector includes the 4 parameterized coordinates of the predicted bounding box, and $m_{i}^{*}$ is a vector of the ground-truth box associated with a positive anchor. $L_{cls}$ is the classification loss to express log loss for two classes (object vs. not object). $L_{reg}(m_{i},m_{i}^{*})=R(m_{i}-m_{i}^{*})$ is the regression loss, where *R* is the robust loss function (smooth L1). The term $p_{i}^{*}L_{reg}$ denotes the regression loss which is activated only for positive anchors ($p_{i}^{*}$ = 1) and is disabled otherwise ($p_{i}^{*}$ = 0). The outputs of the *cls* and *reg* layers consist of $\left\{p_{i}\right\}$ and $\left\{m_{i}\right\}$, respectively. The two terms are normalized with Ncls, Nreg, and a balancing weight λ.

Some RPN proposals highly overlap with each other. To reduce redundancy, NMS, on the proposal regions based on their cls scores, is adopted. The IoU threshold for NMS is set 0.7, which leaves us about 2k proposal regions per image. NMS does not harm the ultimate detection accuracy, but substantially reduces the number of proposals. After NMS, the top-N ranked proposal regions are used for further detection and segmentation.

The process of bounding box generation based on RPN is shown in Figure 5. Figure 5a shows the source image, Figure 5b shows the results that the anchors are targets or non-targets; Figure 5c shows the results of the coordinate corrections on the target anchors.

### 3.4. The Three Branches of Detection and Segmentation

The MS R-CNN outputs the three branches, including performing the proposal classification, regression, and a binary mask for each RoI in parallel. The first two branch structures use two fully connected (FC) layers for the region of interest (RoI) performing classification prediction and BB regression. The last branch is instance segmentation, which uses a fully convolutional network (FCN) [18] for predicting a mask from each BB.

Formally, during training, a multi-task loss ($L_{\mathrm{total}}$) on each BB is defined:(3)$$L_{\mathrm{total}}=L_{box}+L_{\mathrm{cls}}+L_{mask}$$
where $L_{box}$ and $L_{\mathrm{cls}}$ are the bounding-box and classification loss, $L_{mask}$ was defined as the average binary cross-entropy loss. For a BB associated with ground-truth class *k*, $L_{mask}$ is only defined on the *k*-th mask in MS R-CNN model, $L_{\mathrm{box}}$, $L_{\mathrm{cls}}$ and $L_{mask}$ are identical as those defined in [23].

The output of the three branches is shown in Figure 6. Figure 6a shows the BB’s positions can be adjusted using a regression network. Figure 6b shows the BBs that are assigned a score for each class label using a classification network after NMS. Figure 6c shows the detection and segmentation final results of group-housed pigs.

### 3.5. Improving Non-Maximum Suppression

For detection tasks, NMS is a necessary component after BB regression, which is a post-processing algorithm for redundancy removal of detection results. It is a handcrafted algorithm, to greedily select high-scoring detections, and remove their overlapping, low confidence neighbors. The NMS algorithm first sorts the proposal boxes according to the classification scores from high to low, then the detection box m with the highest score($s_{m}$) is selected, and the other boxes with obvious overlap (Intersection over Union (IoU) > threshold used *N*_t_) are suppressed. This process is recursive until all proposal boxes are traversed. Traditional NMS processing methods can be expressed by the following fraction resetting function:(4)$$s_{i}=\{{}_{0,iou(b_{m},b_{i})>=N_{t}}^{s_{i},iou(b_{m},b_{i})<N_{t}}$$
where $b_{m}$, $b_{i}$, $s_{i}$ and $N_{t}$ denote the detection box with the maximum score, the *i*-th of detection box, the score of the *i*-th detection box, and threshold value. NMS sets a hard threshold $N_{t}$ while deciding what should be kept or removed from the neighborhood of $b_{m}$.

NMS performs well in generic object detection, adopting different thresholds; however, due to local maximum suppression, there is missing detection for the easily overlapped target objects. In the group-housed pig-breeding environment, there are serious adhesions and high overlaps in group pigs. To improve the detection performance of group-housed pigs, an improved NMS algorithm named soft-NMS [24] is used. 

The soft-NMS algorithm process is as follows in Figure 7:

where $f(IoU(b_{m},b_{i}))$ is the overlap based weighting function with a Gaussian penalty function in soft-NMS as follows,
(5)$$f\left(\mathrm{IoU}\left(b_{m},b_{i}\right)\right)=e^{-\frac{IoU{\left(b_{m},b_{i}\right)}^{2}}{\sigma }},\forall b_{i}\notin D$$

This rule is applied in the algorithm for each iteration and scores of all remaining detection boxes are updated. In the soft-NMS algorithm, the computational complexity of each step is O(N), where *N* is the number of detection boxes. Moreover, the algorithm updates the scores for all detection boxes that overlap with $b_{m}$. Therefore, for *N* detection boxes, the computational complexity of soft-NMS is $O(N^{2})$, and it is the same as traditional NMS. Soft-NMS hence does not require any additional training and uses the same running time with NMS’s detectors.

### 3.6. Experiment Detail

The experimental environment is described below as follows: 

PC: CPU, Intel^®^ Xeon(R) CPU E5-2620 v4 @ 2.10 GHz × 8; Memory, 64 GB; Graphics, Tesla K40c.

OS: Ubuntu 16.04, CUDA10.1, Python3, Pytorch1.0, PyCharm, Jupyter Notebook.

The procedure of the PigMS R-CNN model mainly involves three steps: image data annotation, model training, and verification, as shown in Figure 8. Firstly, the training set and test set are labeled via the annotation tool, and the corresponding annotation files with JSON format are obtained. The train images, test images, and the corresponding annotation JSON files form the pig object dataset for training. Then, the pig.py of the training file adopts the PigMS R-CNN algorithm to set up an instance segmentation model using python3.6 on the pig object dataset, which includes 290 training and 130 test images. Finally, the validated pig file inspect_pig_model.ipynb written in jupyter notebook software is used to test set and get the pig object detection and segmentation result by calling the built model file of pig.py.

Back-propagation and stochastic gradient descent (SGD) are used to train the PigMS R-CNN model. For RPN networks, each mini batch comes from a single image containing many positive and negative example anchors. The input image is resized to 800 pixels on its shorter side. Synchronized SGD is used to train the model on eight GPUs. Each mini batch includes two images per GPU and 512 anchors in each image with a weight decay of 0.001 and a momentum of 0.9. The learning rate is 0.01 for the first 6000 mini-batches and 0.001 for the next 1000, and 0.0001 for the next 1000. NMS is performed over the proposals with an IoU threshold of 0.7. $N_{t}$ and σ in soft-NMS are set to 0.7 and 0.5, respectively. The pre-trained model of ResNet-101 was available at https://dl.fbaipublicfiles.com/detectron/ImageNetPretrained/MSRA/R-101.pkl (accessed on 7 May 2021). The experimental results, models, and images are obtained by the Baidu network disk’s address (https://pan.baidu.com/s/1_BOpAJ8trdjhBZO1fe4hWA, accessed on 23 April 2021), where the extracted code is a5h6. The precision–recall curve with the corresponding F1 score is used as the evaluation metric for pig detection [23]. The F1 score is computed as: (6)$$F1=\frac{2*Precision*Recall}{Precison+Recall}$$

## 4. Results and Discussion

### 4.1. Experimental Results of the MS R-CNN Model

We verified the reliability and effectiveness of the algorithm using the test dataset, which includes 130 images with 1920 × 1080 pixels. The precision–recall curve (PRC) of MS R-CNN is shown in Figure 9. The black dotted line expresses the points where recall and precision are identical. The more convex the purple line PRC is, the better the result is. The results of the red line PRC show that this method is suitable for pig detection and segmentation in a group-housed pig environment.

The pig segmentation recall, and precision, are also shown in Table 1. Among 130 images, 50 images were taken from the front view images, including 352 pig objects. Moreover, 80 images were taken from the top view images, including 795 pig objects. Among a total of 1147 test pig objects, there were 1159 pigs detected, and 1064 pigs were correctly detected. The MS R-CNN model achieved a recall(R) rate of 92.76%, a precision (P) rate of 91.80%, and an F1 value of 0.9228. In the front view, 296 pig objects were correctly detected among a total of 369 pigs, the recall rate was 84.09%, and the precision rate was 80.22%. In the top view, 768 pig objects were correctly detected among a total of 790 pigs, the recall rate was 96.60%, and the precision rate was 97.22%. The average detection time for each image was 0.5233 s—favorable for the actual production requirements.

As can be seen from Table 1, the result of the top view are significantly better than that of the front view. There are two main reasons: (1) by comparison with the top view, the overlap and adhesion between pigs in the front view were more serious. At the same time, the morphology and features were more complicated. (2) The number of verified images in the front view was close to that from top view. However, due to the limited range from the front view and serious occlusion of pigs, the number of pig objects in the front view was far lower than that in top view, accounting for only 30.69% of the total number of pig objects. Due to the influence of these two factors, the detection results of pig objects in the front view was lower than that of pig objects from the top view.

Figure 10 shows some experimental results of the detection and segmentation of pig objects. The dotted lines in the figure indicate all pig objects detected; the yellow solid line is marked as a pig object detected by mistake, and the red solid line is marked as a pig object detected by omission. The MS R-CNN model achieved good detection and segmentation accuracy for pig objects. However, it had some defects, mainly due to imprecise segmentation of the mask, incomplete detection of pig ears, legs, and tails, and a few cases of false or missed detection.

The results of the top and front view image detections are analyzed below. For the front view images, the main reasons for failure in the detection results are as follows: (1) the front view images included small pig objects, which caused false and missed detection. (2) The front view images included an overlap of pig bodies and disturbance of the light shadow, resulting in missed pigs (as shown with a solid line red box in Figure 10f,h, and false detection of the pig objects (as shown with the solid line yellow box in Figure 10f,h. For the top view images, the main reason for failure in detection and segmentation results was the serious adhesion of the pigs. If the distance between pigs in group-housed pigs is relatively close, it is prone to missed pig objects (as shown in the solid line red box in Figure 10d).

In addition, for the missed detection caused by adhesion, the results of the top view images are more serious than that of the front view images. The missed detection in two pig objects caused from top view images can be divided into two cases, where the two pigs were mistaken as a pig object (solid line orange box marked in Figure 11); one pig was correctly detected while another pig was missed, as shown with the solid line blue box marked in Figure 11.

### 4.2. The Instance Segmentation and Detection Result of the PigMS R-CNN

To improve the missed and wrong detection of target pigs caused by overlapping, adhesion, and other complex environmental issues in a crowded room, the paper used the soft-NMS method instead of the traditional NMS in the MS R-CNN model, without adding extra times. The same 130 images of the test were used for validation, using soft-NMS with a threshold value of 0.5 and a traditional NMS algorithm.

The results of PigMS R-CNN are illustrated in Table 2. Among a total of 1147 test pig objects, 1138 pigs were detected, and 1071 pigs were correctly detected. The PigMS R-CNN model achieved a recall rate of 93.37% and a precision rate of 94.11%, which is better than the MS R-CNN model (Table 2, columns 6 and 7). The PigMS R-CNN based on soft-NMS leads to less missed and incorrectly detected objects.

### 4.3. Discussion

A PigMS R-CNN method is developed for group-housed pig detection and instance segmentation under the natural scene. Previous studies demonstrated that the MS R-CNN method can achieve good detection accuracies with a recall rate of 0.9276 and a precision rate of 0.918. However, the method would not work well to separate the touching-pigs under the overlapping backgrounds of group-housed pigs. To solve the problems, the improved approach was developed by soft-NMS. 

First, the comparison of the touching-pigs detection between the MS R-CNN method with NMS and soft-NMS is discussed. Figure 12 presents the comparisons of some examples of detection results between NMS and soft-NMS, with the same threshold of 0.5. The different solid lines in the figure indicate all pig objects detected, the red dotted line is marked as the pig object, detected successfully with the red arrows, and the blue arrows point to the missed detection.

The MS R-CNN method with NMS and soft-NMS can successfully detect the group-housed pig under a “non-close together: environment. However, the MS R-CNN method with NMS (e.g., Figure 12 left column) produces serious missed detection results under dense touching-pig conditions. For example, the pigs in Figure 12a,c,d are too close together to distinguish individual pigs. The reason is that NMS selects the BB with the max score and sets the scores for neighboring detections to zero. Therefore, if an object actually existed in the overlap threshold, it would be missed, and this would reduce the recall rate. By contrast, the soft-NMS algorithm can overcome the influence of the connected pigs by decaying the scores of neighboring detection boxes, which have an overlap by a Gaussian penalty function, and obtain reliable pig detection results (e.g., Figure 12 right column). The soft-NMS method proved significantly effective at improving the missed detection of touching-pigs.

Table 3 shows the comparison of results between MS R-CNN and PigMS R-CNN when the IoU thresholds were set at different values (0, 0.3, 0.5, 0.8, and 0.9). It is observed that the PigMS R-CNN model is stable with an average recall of 94% and average precision of 93%. By contrast, the MS R-CNN model achieves an average recall of 93.5% and average precision of 90%. In particular, when the IoU threshold value is set to zero, the precision of MS R-CNN is 86.33%, which is lowest in other IoU threshold value situations, and that of the PigMS R-CNN, is 92.77%. Therefore, PigMS R-CNN can obtain better results than MS R-CNN in precision when the two methods achieve almost equal the recall rates.

Next, some detection errors of the soft-NMS algorithm are analyzed and given in Figure 13. In our experiments, most of the false detections occur under two situations. First, false detections can occur when pigs from top view show incomplete appearances and severe shape deformation when compared with the pigs in the training images, such as the pigs show part appearances, as shown with the green box in Figure 13a. Second, for pigs from the front view, due to the heavy overlapping, false detections in the soft-NMS algorithm can occur, as shown in Figure 13b. Therefore, inferior environments will reduce the detection performance for front view images. To overcome these limitations, more advanced detection and segmentation techniques can be used to improve image quality. A pig key-point detection algorithm based on human posture [27,28] can be studied and utilized to enhance the reliability of the detection results.

## 5. Conclusions and Future Work

The group-housed instance segmentation of pigs in a natural environment is a significant operation to efficiently manage pig farms. However, using a traditional method, group-housed pigs cannot be separated accurately in real-time for heavily overlapped pigs in complex backgrounds. 

In this paper, an improved MS R-CNN framework with soft-NMS was proposed to obtain the locations and segmentation of each pig in group-housed pig images. To prevent the missed and wrong detection of target pigs, caused by overlapping, adhesion, and other complex environmental issues in a crowded room, this paper employs the soft-NMS method instead of the traditional NMS in the MS R-CNN model, without adding extra times. All boxes, which have an overlap greater than a threshold $N_{t}$ in traditional NMS, are given a zero score. Compared with NMS, soft-NMS rescores neighboring boxes instead of suppressing them altogether, which obtains improvement in precision and recall values. 

Based on the pig detection and segmentation results for 130 images, with 1147 for top view and front view, the basic MS R-CNN framework obtained results with an F1 of 0.9228, while the target pigs using PigMS R-CNN had an F1 of 0.9374 in complex scenes. This algorithm can achieve good performance in terms of F1 without adding extra time. Our work on pig instance segmentation supports the foundation for pig behavior monitoring, posture recognition, and other related applications, such as body size and weight measurement estimations of pigs. Furthermore, it provides a deep learning framework for detecting and segmenting animals using overhead and front view cameras in a natural environment. In the future, we will develop pig behavior applications, such as pig monitoring of drinking water and fighting behavior under group-housed pigs’ scenes.

## Figures and Tables

**Figure 1 sensors-21-03251-f001:**
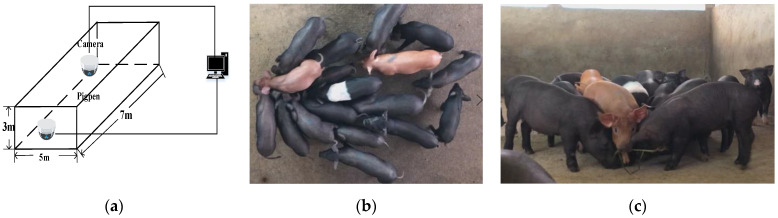
(**a**) Video surveillance system for data acquisition, (**b**) top view image, and (**c**) front view image.

**Figure 2 sensors-21-03251-f002:**
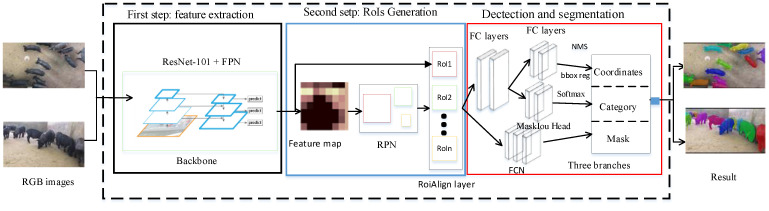
Flowchart of the detection and segmentation algorithm based on PigMS R-CNN.

**Figure 3 sensors-21-03251-f003:**
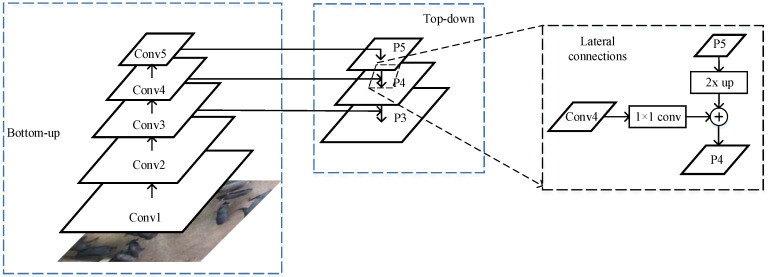
The architecture of feature pyramid network.

**Figure 4 sensors-21-03251-f004:**
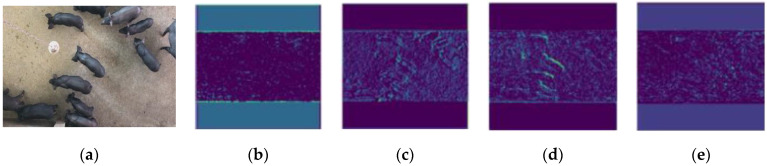
The feature map of backbone architecture based on ResNe101+FPN. (**a**) shows source image, (**b**–**d**) extract low-level features, and (**e**) represents high-level abstract features.

**Figure 5 sensors-21-03251-f005:**
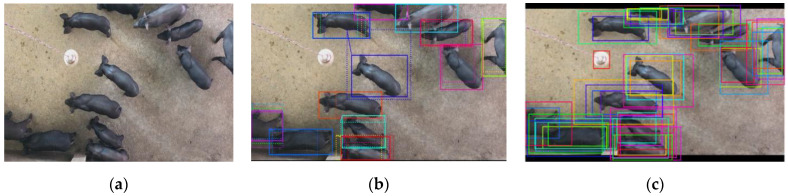
The process of bounding box generation based on RPN, (**a**) shows the source image, (**b**) shows the results that the anchors are targets or non-targets, and (**c**) shows the results of the coordinate corrections on the target anchors.

**Figure 6 sensors-21-03251-f006:**
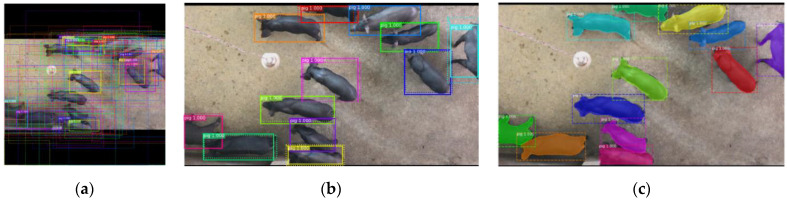
The output of three branches. (**a**) The BB positions, which can be updated using a regression network, (**b**) the classification prediction of each BB, (**c**) the detection and segmentation mask result of each BB.

**Figure 7 sensors-21-03251-f007:**
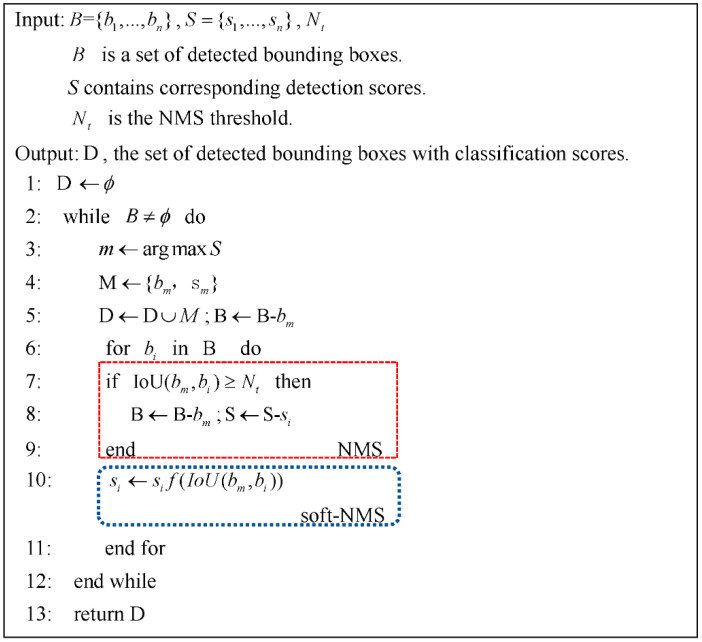
The soft-NMS algorithm.

**Figure 8 sensors-21-03251-f008:**
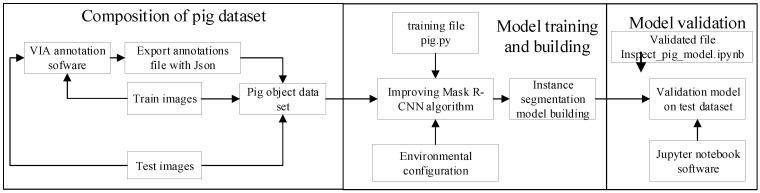
Establishment process of the pig instance segmentation model based on the PigMS R-CNN.

**Figure 9 sensors-21-03251-f009:**
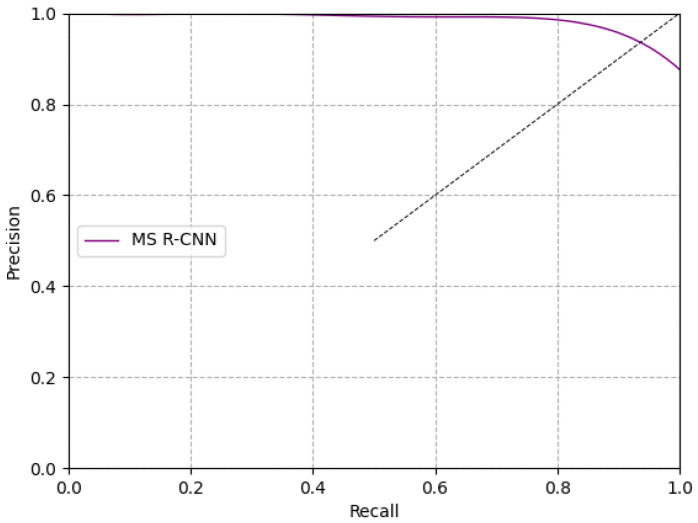
Precision–recall curves based on MS R-CNN.

**Figure 10 sensors-21-03251-f010:**
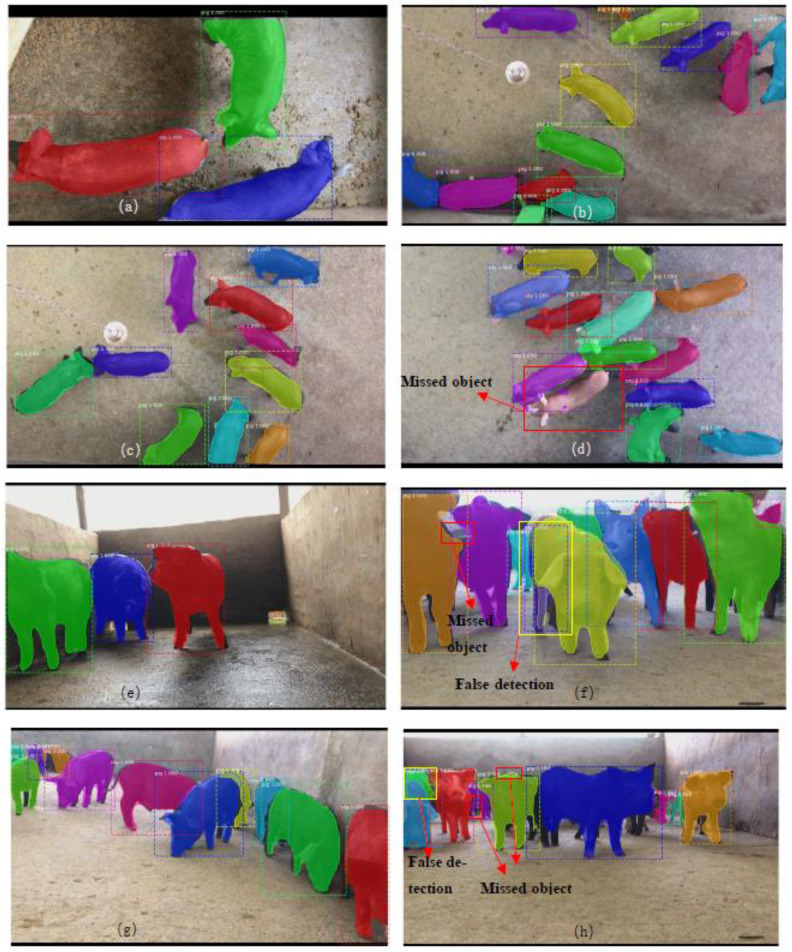
Detection and instance segmentation of the pig objects. Results include a small number of pigs (**a**), a medium number of pigs (**b**,**c**), a large number of pigs (**d**) in the top view images. Results include a small number of pigs (**e**), a medium number of pigs (**f**,**g**), a large number of pigs (**h**) in the front view images.

**Figure 11 sensors-21-03251-f011:**
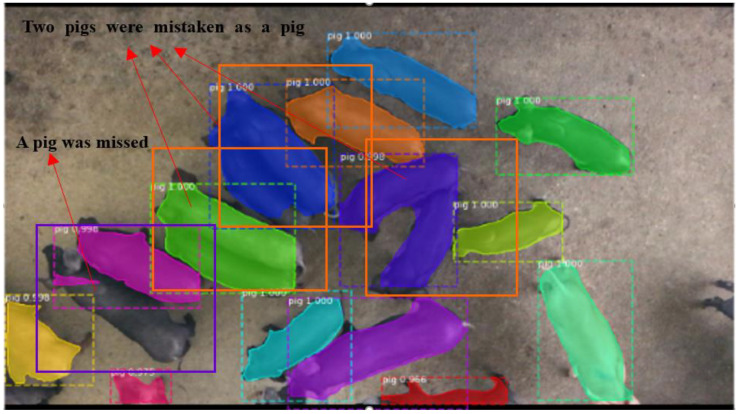
The results of missed detection from the top view image.

**Figure 12 sensors-21-03251-f012:**
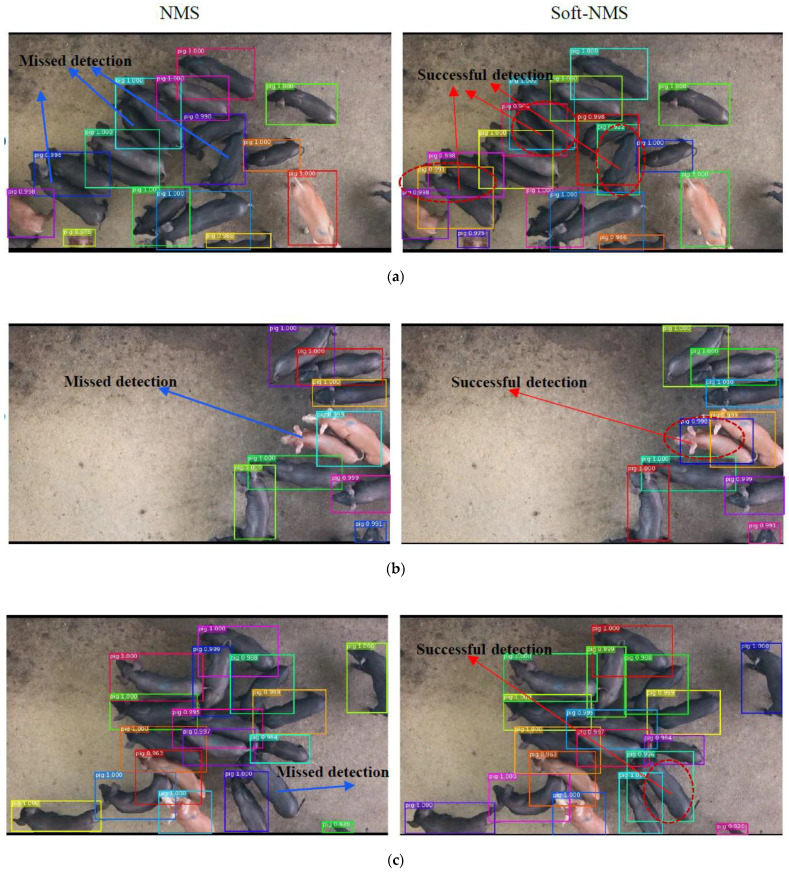
The comparison of detection between NMS and soft-NMS. The left column of (**a**–**d**) shows that the MS R-CNN method with NMS produces serious missed detection results under dense touching-pig conditions. The right column of (**a**–**d**) shows that the PigMS R-CNN approach with the soft-NMS algorithm overcomes the influence of the connected pigs by a Gaussian penalty function.

**Figure 13 sensors-21-03251-f013:**
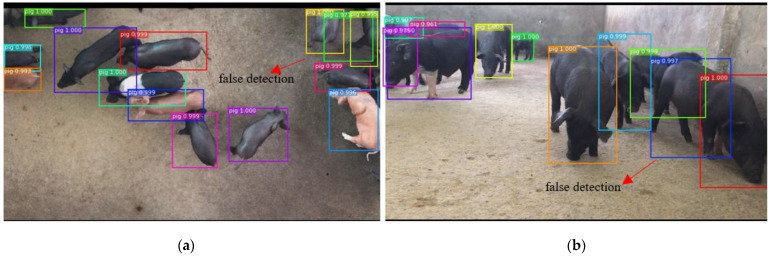
Example of detection failures of the proposed algorithm. (**a**) shows that false detections can occur when pigs from top view show incomplete appearances and severe shape deformation. For pigs from the front view, (**b**) shows that due to the heavy overlapping, false detections in the soft-NMS algorithm can occur.

**Table 1 sensors-21-03251-t001:** The detection results of 130 images based on MS R-CNN.

Image Type	The Total of Pig Objects	The Detected Number	The Correct Detected Number	Recall (%)	Precision (%)	F1
Front view	352	369	296	84.09	80.22	0.8211
Top view	795	790	768	96.60	97.22	0.9691
Total number	1147	1159	1064	92.76	91.80	0.9228

**Table 2 sensors-21-03251-t002:** Detection and instance segmentation result of PigMS R-CNN.

Image Type	The Total of Pig Objects	The Detected Number	The Correct Detected Number	Recall (%)	Precision (%)	F1
Front view	352	350	294	83.52	84.00	0.8376
Top view	395	788	777	97.74	98.60	0.9817
Total number	1147	1138	1071	93.37	94.11	0.9374

**Table 3 sensors-21-03251-t003:** The comparison of results of different IoU thresholds between MS R-CNN and PigMS R-CNN.

Method	0			0.3			0.5			0.8		
	R	P	F1	R	P	F1	R	P	F1	R	P	F1
MS R-CNN	94.68%	86.33%	0.9031	93.55%	91.59%	0.9256	93.27%	90.19%	0.9170	92.68%	92.6%	0.9264
PigMS R-CNN	94.85%	92.77%	0.9380	93.85%	92.99%	0.9342	93.95%	92.69%	0.9332	93.33%	93.80%	0.9356

## Data Availability

Not applicable.

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
