# Peer review of "Automatic Detection and Segmentation for Group-Housed Pigs Based on PigMS R-CNN†"

_sensors, 2021, doi:10.3390/s21093251_

Round 1

Reviewer 1 Report

The authors presented a pig detection algorithm using R-CNN. A top and lateral views are considered in the study.

The study suggests a new soft-NMS that improves the detection accuracy. It seems that this was the contribution of the paper.

In this case, a formal explanation should be given to support the conclusions and clarify the obtained results.

The authors should sketch in a figure the advantage of using such weighting function and show in which cases it is better than the traditional NMS. A set of cases with simple IOU configurations will be clearer and easy to figure out the main contribution.

In Table 3, the best results (i.e. 94.68%) should be given by an IoU threshold value of 0.7 as specified in the text? but the table shows a corresponding threshold of 0. ?

In figure 7.: The edged blue block and the red one should be correctly  indented

Author Response

Please see the attachment about the detailed response file.

The authors presented a pig detection algorithm using R-CNN. A top and lateral views are considered in the study.

Responses: Yes, we presented a pig detection algorithm from the top and lateral views, which can detect pigs more comprehensively than the single view.

The study suggests a new soft-NMS that improves the detection accuracy. It seems that this was the contribution of the paper.

In this case, a formal explanation should be given to support the conclusions and clarify the obtained results.

Responses: We used soft-NMS to improve the detection accuracy, which was our main contribution. In Section 4.2 and 4.3, First, we use the same dataset and super-parameters to complete the experiments and compare the results of the two methods (as shown in Table 3). Then, we analyze the reason why soft NMS is superior to the traditional NMS, and compare it with the detection result chart, as shown in Figure 12. Therefore, we have given a formal explanation to support the conclusions and clarify the obtained results.

The authors should sketch in a figure the advantage of using such weighting function and show in which cases it is better than the traditional NMS. A set of cases with simple IOU configurations will be clearer and easy to figure out the main contribution.

Responses: Thank you for your good advice. In fact, our idea is the same as yours. In Table 3, We use different thresholds to show that soft NMS is superior to traditional NMS in the precision and recall rate under dense touching-pig conditions.

In Table 3, the best results (i.e. 94.68%) should be given by an IoU threshold value of 0.7 as specified in the text? but the table shows a corresponding threshold of 0. ?

Responses: In Mask R-CNN’s experiments, an IoU threshold value of 0.7 is given as the default. In table 1, MS R-CNN using default IoU threshold value of 0.7 can achieve the results with the precision rate of 91.80% and the recall rate with 92.76%. And in Table 3, the recall rate with 94.68% is not the best results. This reason is that the precision rate is only 86.33%. For experimental results, we need to consider both the precision and recall rate. To express the performance more clearly, we add the value of evaluation parameter F1 in Table 1,Table2,and Table 3.

In figure 7.: The edged blue block and the red one should be correctly  indented

Responses: Thanks for reviewer’s suggestion. In the source manuscript, Figure 7 is shown as normal (as shown in Fig.1). This may be caused by the editing when processing the image format (as shown in Fig.2). We will correct this error by the editor in Fig.7.

Reviewer 2 Report

It seems that the proposed method applies only to the image of pigs. According to the abstract, the significance of the proposed method concerns only the exploration for pigs tracking and monitoring. In this case, the importance of this method is limited, even to the field of image processing.

Some statements and terms of this manuscript are somewhat misleading. For example, the term “accurate and reliable instance segmentation for adhesive pigs” misleads readers into thinking that some actions are performed on pigs, rather than pigs’ images.

The logic of the first sentence of the manuscript is quite strange to me. Why is that monitoring the welfare and health of pigs increasingly important?

The experiment is not convincing because no benchmark method was involved. I would suggest a major revision on performing a comparative study between the proposed method and the state-of-the-art methods.

Author Response

It seems that the proposed method applies only to the image of pigs. According to the abstract, the significance of the proposed method concerns only the exploration for pigs tracking and monitoring. In this case, the importance of this method is limited, even to the field of image processing.

Responses: We mainly use computer vision to detect and segment pigs. According to the abstract, the significance of the proposed method concerns the exploration for pigs tracking and monitoring which can find abnormal conditions of pigs and ways to avoid loss especially when pigs are sick. Therefore, this method is very important for video automatic monitoring of pig behavior.

Some statements and terms of this manuscript are somewhat misleading. For example, the term “accurate and reliable instance segmentation for adhesive pigs” misleads readers into thinking that some actions are performed on pigs, rather than pigs’ images.

 Responses: Thank you for your good advice. We have read the whole paper carefully and revised these problems. We have also invited a couple of native English speakers to revise the manuscript. I hope the manuscript looks better.

The logic of the first sentence of the manuscript is quite strange to me. Why is that monitoring the welfare and health of pigs increasingly important?

 Responses: We are very sorry that the logic of the first sentence makes you feel strange.

Videos monitoring pig technology offers multiple advantages for monitoring key indicators of welfare in pigs that can help to address new challenges in the pig sector. This technology allows continuous and objective monitoring on individual and group levels, on multiple farms, and also in various environments. Real-time implementation of videos-based monitoring systems makes it possible to detect a problem and to take immediate action to improve the life of the animal under consideration. So, we think that the applications of videos monitoring pig welfare and health have become more and more important in the modern pig industry.

The experiment is not convincing because no benchmark method was involved. I would suggest a major revision on performing a comparative study between the proposed method and the state-of-the-art methods.

Responses: In the task of instance segmentation, the Mask R-CNN method proposed in 2017 is one of the most commonly used benchmark methods. Mask scoring R-CNN (MS R-CNN) is an improved version of mask RCNN two years later, which contains a network block to learn the quality of the predicted instance masks.

In our conference paper (Instance Segmentation Based on Mask Scoring R-CNN for Group-housed Pigs), we perform a comparative study between MS R-CNN and Mask R-CNN methods (as shown in Fig.3). The more convex the red line PRC is, the better the result is. Compared with Mask R-CNN model (Blue dotted line), the Mask Scoring R-CNN (Red solid line) can achieve better results.

In this manuscript, based on MS R-CNN, we proposed the PigMS R-CNN method. The results of MS R-CNN method are shown in Table1, and the results of the PigMS R-CNN are listed in Table2. The results’ comparisons between PigMS R-CNN and MS R-CNN methods are shown in table 3.  

    To sum up, we have performed a comparative study between the proposed method and the state-of-the-art methods. These experiments are convincing enough. Our experimental results, models and images are obtained by the address ( https://pan.baidu.com/s/1_BOpAJ8trdjhBZO1fe4hWA) where the extracted code is a5h6.

 Please see the attachment about the detailed response file.

Round 2

Reviewer 1 Report

Thanks to the authors for their responses.

However, there are still some English writing errors (e.g.   “which using a fully” I guess that the correct form should be “which uses a fully”. The authors should check for other errors.

Also, I recommend removing the figure of the soft-NMS algorithm and only put the reference [24]. Since this is not the author contribution.

Author Response

Thanks to the authors for their responses.

However, there are still some English writing errors (e.g.   “which using a fully” I guess that the correct form should be “which uses a fully”. The authors should check for other errors.

Responses: We have read the whole paper carefully and revised these English writing errors. We have also invited a couple of native English speakers to revise the manuscript. We hope that the manuscript can meet the requirements of the journal.

Also, I recommend removing the figure of the soft-NMS algorithm and only put the reference [24]. Since this is not the author contribution.

Responses: The soft-NMS algorithm in our manuscript is a significant role in improving pig’s detection performance under group-housed pig’s environment, and we have detailed compared the results of MS R-CNN and PigMS R-CNN for using the soft-NMS algorithm in Table 3. To make readers better understand the role of the soft-NMS algorithm, we consider that this paper should explain the detailed working process of this algorithm.

Reviewer 2 Report

The authors have addressed most of my comments except for the first one, namely the significance of this paper. In my last round of review, I commented that “It seems that the proposed method applies only to the image of pigs. According to the abstract, the significance of the proposed method concerns only the exploration for pigs tracking and monitoring. In this case, the importance of this method is limited, even to the field of image processing.” The authors responded with some words, but these words did not address my concern.
